# Quantitative pathogenicity and host adaptation in a fungal plant pathogen revealed by whole-genome sequencing

Reda Amezrou [1] ✉, Aurélie Ducasse[1], Jérôme Compain[2], Nicolas Lapalu [1,2], Anais Pitarch[1], Laetitia Dupont[1], Johann Confais[1], Henriette Goyeau[1], Gert H. J. Kema [3], Daniel Croll [4], Joëlle Amselem[2], Andrea Sanchez-Vallet [5] & Thierry C. Marcel [1] ✉

Knowledge of genetic determinism and evolutionary dynamics mediating host-pathogen interactions is essential to manage fungal plant diseases. Studies on the genetic architecture of fungal pathogenicity often focus on large-effect effector genes triggering strong, qualitative resistance. It is not clear how this translates to predominately quantitative interactions. Here, we use the *Zymoseptoria tritici*-wheat model to elucidate the genetic architecture of quantitative pathogenicity and mechanisms mediating host adaptation. With a multi-host genome-wide association study, we identify 19 high-confidence candidate genes associated with quantitative pathogenicity. Analysis of genetic diversity reveals that sequence polymorphism is the main evolutionary process mediating differences in quantitative pathogenicity, a process that is likely facilitated by genetic recombination and transposable element dynamics. Finally, we use functional approaches to confirm the role of an effector-like gene and a methyltransferase in phenotypic variation. This study highlights the complex genetic architecture of quantitative pathogenicity, extensive diversifying selection and plausible mechanisms facilitating pathogen adaptation.

Fungal diseases cause major damage to crop production and threaten food security worldwide[1]. Understanding the molecular dialogue between pathogens and their hosts is essential to design durable and effective control strategies. Major molecular factors in the ability of fungal pathogens to cause disease are effectors. Those are proteins delivered into the host apoplast or translocated inside cells to manipulate host physiology or suppress its immunity to favor infection[2]. In turn, plants respond to pathogen invasion by activating defence mechanisms based on recognition of pathogen-associated molecular patterns and effectors by cell surface and intracellular receptors. The discovery of resistance genes (*R*) that encode receptors

has provided potential means to control fungal diseases through resistance breeding[3]. However, pathogens have repeatedly overcome host resistance by evading recognition or by suppression of host immunity. These rapid adaptations are mainly driven by genetic variation at pathogenicity genes and genome evolution[4]. Gene diversification, deletions or horizontal acquisitions have generated adaptive variants at loci encoding effectors. Transposable elements (TEs), high mutation and recombination rates are thought to contribute considerably to the extensive genome variation in many fungal species[5–8].

In the case of antagonistic host-pathogen interactions, the fate of genetic variation is determined partly by the selection pressure

[1]Université Paris-Saclay, INRAE, UR BIOGER, Palaiseau, France. [2]Université Paris-Saclay, INRAE, UR URGI, Versailles, France. [3]Plant Research International B.V., Wageningen, The Netherlands. [4]Department of Ecology and Evolution, Université de Neuchâtel, Neuchâtel, Switzerland. [5]CBGP, INIA, Campus de Montegancedo UPM, Pozuelo de Alarcón, Madrid, Spain. ✉e-mail: reddamez@gmail.com; thierry.marcel@inrae.fr

exerted by host *R* proteins. This is especially true when *R* genes and effectors are involved in a gene-for-gene (GFG) relationship[9–12]. Moreover, the diversity of host genotypes in *R*-gene content exerts heterogeneous selection pressure on pathogen populations, giving rise to the co-maintenance of multiple pathogen variants better adapted to specific subsets of hosts[3,13]. Hence, in addition to the identification of effector genes, elucidating the mechanisms contributing to gene diversification provides critical information about their evolvability. This knowledge has tremendous applications for the durability of crop protection strategies[14].

The first fungal effectors to be identified were small secreted proteins (SSP). SSPs trigger strong resistance (i.e. encoded by avirulence genes) as a result of their recognition by plant immune receptors following a typical GFG relationship[15]. The increasing availability of fungal genomes over the past decade allowed significant advances in investigating molecular plant–fungal interactions and facilitated effector discovery. However, effector biology has focused almost exclusively on SSPs despite the growing evidence that other genetic factors are also involved in host colonization[16]. For instance, small noncoding RNAs and fungal secondary metabolites play similar effector roles in various plant pathogenic fungi[17]. Other fungal proteins were shown to interact with the host immune system or act to promote pathogenicity; these include for example cell wall–degrading enzymes[18], protease inhibitors[19] or disruptors of the hormone-signaling pathway[20].

Effector discovery also led to the systematic search for host genes conferring resistance[15]. Resistance to many fungal diseases can be divided into two categories: (1) qualitative (or race-specific) resistance, which is usually controlled by large-effect genes and is effective against avirulent strains; and (2) quantitative (or non–race-specific) resistance, usually controlled by genes with small-to-moderate effects and is thought to be less specific. While cloning plant immune receptors remains challenging, the discovery of the interacting fungal effectors is considered more straightforward[14]. However, fungal pathogenicity studies often focus on large-effect genes, whose interaction with genetic factors from the host trigger a strong immune response, leading to qualitative resistance. It remains unclear how these findings translate to quantitative models, where the interaction of quantitative components of pathogenicity (also called aggressiveness in the plant pathology literature) and plant resistance, trigger a partial/incomplete immune response. While the quantitative components of pathogenicity are frequently measured in various plant-pathogen interactions[21], there have been only few reports on the genetic basis of quantitative pathogenicity in pathogens and the process of adaptation to quantitative host resistance[22,23]. Thus, elucidating the biology of quantitative pathogenicity factors and their evolutionary dynamics is timely and may require the adoption of complementary experimental approaches. To address this, we used the *Zymoseptoria tritici*–wheat pathosystem, which is known to be largely quantitative *in natura*.

*Zymoseptoria tritici* is an ascomycete filamentous fungus that causes the Septoria tritici blotch (STB) disease on wheat. *STB* is a major threat to wheat crops in Europe and in temperate regions around the world[24]. *Z. tritici* is hemibiotrophic and remains exclusively apoplastic through its infection cycle, which is characterized by an asymptomatic phase of 7 to 14 days, followed by an induction of leaf necrosis[25]. This stage is accompanied by the development of pycnidia containing asexual spores that further spread the disease[26,27]. To date, three *Z. tritici* pathogenicity genes that encode avirulence factors have been cloned. These include *AvrStb6* and *AvrStb9* that are recognized by the *R* genes *Stb6* and *Stb9*, respectively, and lead to a strong qualitative resistance response[12,28,29], and *Avr3D1* which triggers quantitative resistance on cultivars harboring *Stb7* or *Stb12*[10]. In addition to the above-mentioned *R* genes, 18 other race-specific *Stb* resistance genes and numerous quantitative resistance loci have been genetically mapped in wheat[30], but only *Stb6* and *Stb16q* have been cloned to date[31,32]. An ideal approach to identify genetic determinants of

complex traits is through Genome-Wide Association Studies (GWAS). GWAS aims to associate sequence polymorphisms to phenotypic variation, and relies on linkage disequilibrium (LD)—the non-random association of alleles at different loci—to identify causal variants. The approach has been used extensively in plants but remains at its infancy on fungal populations[33,34].

The aim of the present work was to characterize the genetic determinants underlying quantitative pathogenicity in *Z. tritici* and to investigate the evolutionary processes that could facilitate pathogen adaptation to host resistances. We hypothesized that quantitative pathogenicity relies mostly on gene-for-gene interactions between host and pathogen, and that the genes involved are not all encoding SSPs but also genetic factors from the non-secretory pathway. To test our hypotheses, we performed an extensive genome wide association study combined with transcriptomics and population genomics. We identified candidate genes putatively mediating quantitative pathogenicity and confirmed the role of two of these genes using functional approaches. Together, our results generate a comprehensive view on the complex genetic architecture of quantitative pathogenicity and pathogen adaptation to host resistances. Beyond this model, our findings may have broad implications for sustainable disease management.

## Results

### Characterization of the fungal collection

To efficiently perform GWAS and reduce confounding effects, it is important to select a population that is genetically and phenotypically diverse with little stratification. Following these guidelines, we focused on a subset of 103 French *Z. tritici* isolates selected from a larger population for their genetic diversity. We sequenced their genomes and generated a total of 247 Gb of sequence data with an average depth of 61x. After quality assessment, we obtained a total of 1,463,638 Single Nucleotide Polymorphisms (SNPs) that we further filtered for Minor Allele Frequency (MAF) and missing rates, resulting in a final set of 718,810 SNPs. Analysis of functional variant effect revealed that 59,318 SNPs (8.25%) induced missense mutations and 2493 (0.3%) resulted in a loss of function such as premature stop codons, loss of stop or start codons (Table 1). As expected, principal component analysis (PCA) using full polymorphism data did not show any genetic structure; the first three PCs explained less than 4% of the total variance (Supplementary Fig. 1). We further confirmed this with STUC-TURE using a subset of evenly distributed SNPs (1 SNP per 5 kb) and like the PCA analysis, Bayesian clustering failed to infer sub-populations (Fig. 1a). Similarly, individual isolates showed very low genetic relatedness (Supplementary Fig. 2). Genome-wide LD decayed ($r^2 < 0.2$) over physical distance at -0.5 kb (Fig. 1b). Given our high SNP density (1 SNP every 55 bp) we expected marker-trait associations to be resolved to the individual gene level. When pathotyped on wheat differential cultivars, isolates showed a wide virulence spectrum depending on the host cultivar, and strong isolate × cultivar interactions (Supplementary

**Table 1 | Functional variants identified in a population of 103 *Z. tritici* isolates**

| Functional variant effect | Count | Percent |
|---|---|---|
| 5′ UTR | 10,431 | 1.45 |
| Gain of start codon | 1084 | 0.15 |
| Loss of start codon | 127 | 0.02 |
| Synonymous coding | 162,938 | 22.67 |
| Non-synonymous coding | 59,318 | 8.25 |
| Intron | 58,217 | 8.10 |
| Gain of stop codon | 1109 | 0.15 |
| Loss of stop codon | 173 | 0.02 |
| 3′ UTR | 8794 | 1.22 |

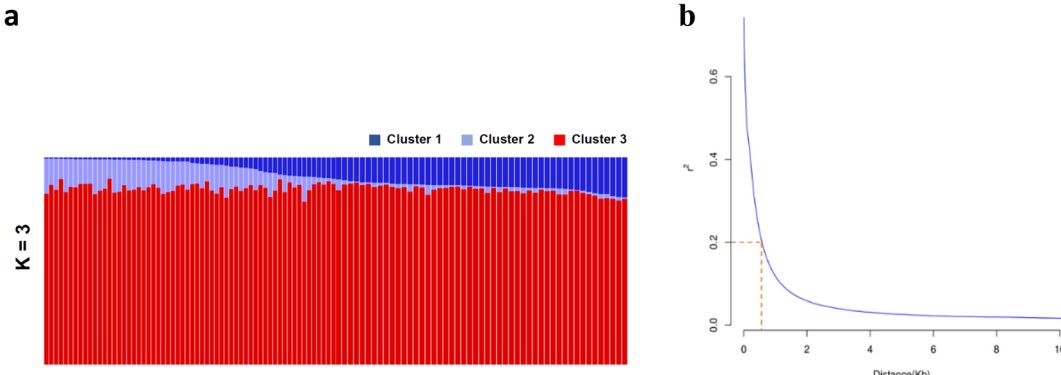

**Fig. 1 | Population structure and linkage disequilibrium (LD) in the _Z. tritici_ population (_n_ = 103). a** Admixture plot of the fungal population. Each vertical bar represents a single isolate and is colored according to the membership coefficient ($Q_i$) to the three sub-population (_K_) clusters identified by STRUCTURE. **b** LD decay over physical distance. The $r^2$ values were calculated between pairs of SNPs up to a physical distance of 20 kb and fitted against the physical distance using a non-linear model. The dotted orange line shows the distance (0.5 kb) at which LD decays at the $r^2 < 0.2$ level.

Tables 1 and 2). Overall, strains isolated from the cultivar "Premio" were more aggressive compared to those isolated from the susceptible cultivar "Apache", even when tested on the cultivar "Apache" (Supplementary Fig. 3).

### A complex genetic architecture and host-specificity underlies pathogenicity

To elucidate the genetic basis of the observed pathogenicity differences, we investigated two key quantitative phenotypes (PLACN and PLACP) using mixed linear models for marker-trait association tests, performed independently on each of the studied cultivars. Collectively, we identified 94 genome-wide significant SNPs and an additional 53 at the 10% False Discovery Rate (FDR) threshold, defining 58 distinct genes (Fig. 2 and Supplementary Data 1). The strongest associations were at the _AvrStb6_ and _AvrStb9_ loci and were detected in all _Stb6_- and _Stb9_- containing cultivars, respectively. Initially not thought to harbor _Stb9_, PLACN on cultivar "Cadenza" was highly associated ($p = 3.57e{-}13$) with polymorphisms in _AvrStb9_, suggesting that the cultivar possesses the corresponding _R_ gene. The presence of _Stb9_ in "Cadenza" was recently confirmed by its tightly linked genetic markers. Interestingly, the single-locus model failed to detect _AvrStb9_ and demonstrates the usefulness of the multi-locus mixed model in detecting loci with larger effects. However, if multiple SNPs were in strong LD, biases may be created while estimating the genetic variance. Nevertheless, the multi-locus approach in our study generally outperformed the traditional single-locus tests (Supplementary Fig. 4). Candidate pathogenicity loci were identified in the thirteen _Z. tritici_ core chromosomes. Despite recent suggestions of accessory chromosomes being enriched in putative effectors[35], we could not map any gene residing on accessory chromosomes in this multi-host GWAS (Fig. 2 and Supplementary Data 1). Only eleven genes were shared among PLACP and PLACN (Fig. 2 and Supplementary Data 1), which was expected, since necrosis coverage and pycnidia production levels were highly variable (Supplementary Fig. 3). Hence, considering both traits in association analyses could reveal genetic determinants influencing both, the conidial production and host susceptibility. The phenotypic variance explained by the identified loci associated with pathogenicity traits on each cultivar ranged from 9.1% to 65.7% (Supplementary Data 1). With the exception of _AvrStb6_ and _AvrStb9_, all of the detected genes accounted for small-to-moderate effects ($R^2 < 30\%$) and similarly, analysis of virulent/avirulent alleles showed quantitative differences (Supplementary Data 1 and Fig. 3b). Further, the number of associated genes varied greatly between cultivars (three candidates for "Bulgaria-88" and "Toisondor" vs. ten for "Israel-493") and ~76% were specific to a single host cultivar (Supplementary Data 1 and Fig. 3a). This weak

overlap observed between genes is a strong indication of host-specificity, in consistence with the different _Stb_ genes or _Stb_ combinations harbored by the cultivar set used here.

### _In planta_ gene expression and functional annotation

To reveal the expression profile of the identified candidate pathogenicity genes using GWAS, we calculated the relative abundance of _Z. tritici_ transcripts, using RNAseq data generated during different _Z. tritici_ infection time courses[25]. Of the 58 GWAS candidates, ten genes exhibited no expression (FPKM = 0) or very low expression (FPKM < 5) throughout the infection cycle, while another ten genes were found to be highly expressed (FPKM above the 95th percentile of the mean expression value) either at specific time points or throughout the infection cycle (Fig. 3c and Supplementary Data 2). Interestingly, _AvrStb6_, _Zt_6_00224_ and _Zt_3_00467_ were among the most highly expressed _Z. tritici_ genes at the disease transitional phase (9 dpi; Supplementary Data 2). Differentially expressed genes _in planta_ were identified by examining the ratio of expression between the Czapek-Dox broth (CDB) culture medium and the five time points of wheat infection. In our set of candidate genes, sixteen were differentially expressed with a FDR below 0.10 and a log2-fold change greater than 2. Among them, 11 were up-regulated at specific time points of the infection (Fig. 3c and Supplementary Data 2). For subsequent investigations, we considered a set of nineteen genes as high-confidence candidates, whose expression profiles supported their potential role in pathogenicity; this included the 16 differentially expressed genes and three additional ones that were highly expressed during the infection (Fig. 3 and Supplementary Data 2).

Eleven protein products out of the nineteen retained as best candidates were predicted to be secreted (Supplementary Table 4). These secreted proteins could be effectors promoting disease or interacting with the cognate _Stb_ resistance genes analyzed in this study. We also found three proteins with predicted transmembrane helices (THMs), a class that is systematically discarded in effector discovery studies. EffectorP predicted seven as putative effectors, with five (excluding _AvrStb6_) lacking homology with other fungal species and might be considered as _Z. tritici_-specific candidate effectors (Supplementary Table 4). The identified conserved protein domains included functions such as hydrolase (e.g., Glycosyl hydrolase), oxidoreductase (e.g., Thioredoxin reductase) or transferase activities (e.g., Methyltransferase; Supplementary Table 4). Using PHIB-Blast, we found that seven candidates shared homology with known pathogenicity factors from other pathogenic fungi and therefore constitute promising candidates mediating quantitative pathogenicity in _Z. tritici_ (Supplementary Table 4).

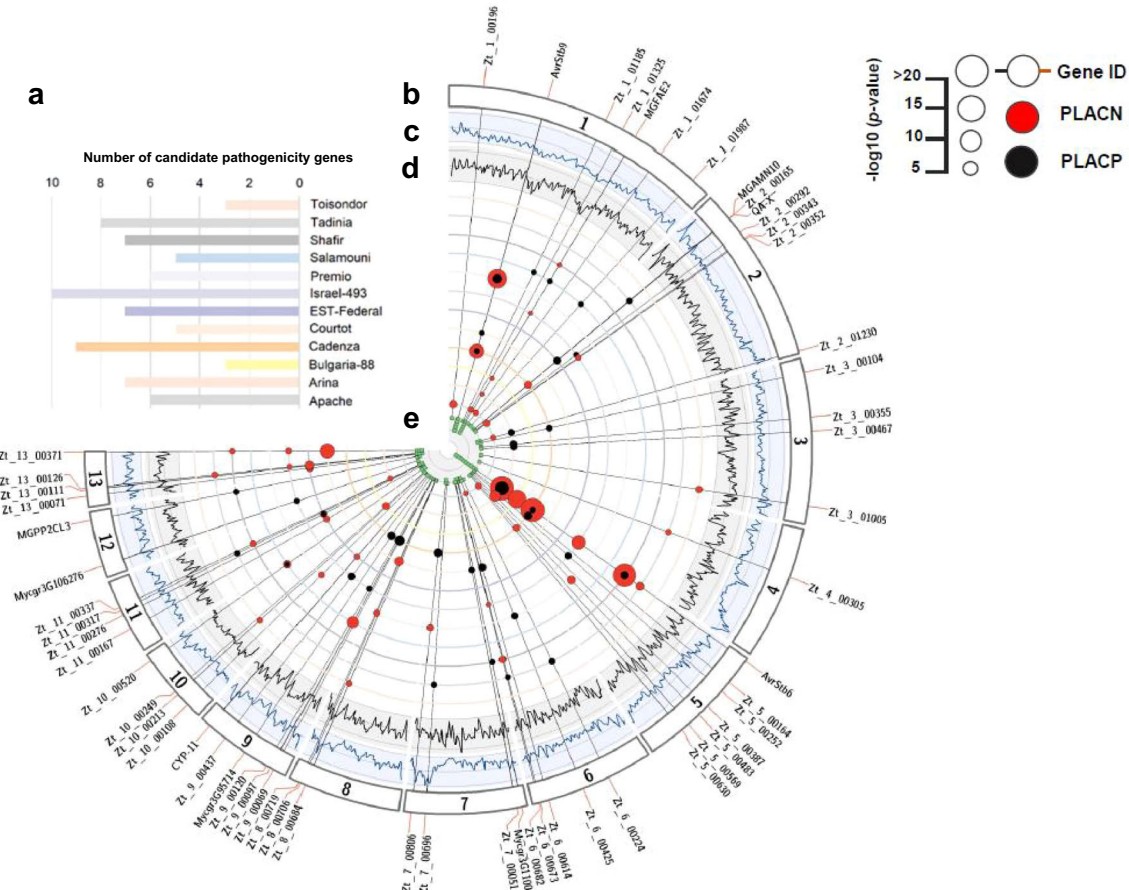

**Fig. 2 | Overview of the identified candidate pathogenicity genes from the multi-host GWAS. a** Number of identified genes per cultivar. **b** Gene IDs and their significant associations. Black dots correspond to PLACP (percentage of leaf area covered by pycnidia) and red dots to PLACN (percentage of leaf area covered by necrosis) marker-trait associations. The larger the dot the stronger the association (*p* value). Radial lines connect dots to the cultivar where it was detected and to the

gene ID. **c** Nucleotide diversity ($\pi$) estimates in a 50 kb sliding window with a step size of 25 kb. Genomic positions are displayed on the *x*-axis and $\pi$ values are displayed on the *y*-axis. The *y*-axis ranges from 0 to 0.045. **d** Tajima's *D* estimates in 50 kb sliding windows with a step size of 25 kb. Genomic positions are displayed on the *x*-axis. Tajima's *D* values are displayed on the *y*-axis. The *y*-axis ranges from −2.5 to 2. **e** The number of associations for each gene (PLACN/PLACP per cultivar).

## Factors of genetic diversity in pathogenicity genes

Gain in virulence and/or escape from host recognition is often mediated by polymorphisms in pathogenicity genes. We investigated sequence variation and diversity in the identified genes and detailed results are summarized in Supplementary Table 5. In particular, we compared patterns of nucleotide diversity in candidate pathogenicity genes with the genome-wide levels. We observed higher genetic diversity in pathogenicity genes compared to other protein-coding genes (median $\pi_{\text{Pathogenicity}} = 0.0212$ vs. $\pi_{\text{Genome-wide}} = 0.0079$; $p < 0.01$; Fig. 4a). We found no significant impact of loss of function mutations or presence/absence polymorphisms, except for *Mycgr3G110052*. In addition, we wanted to examine which mechanisms were likely to contribute to the high sequence diversity in our candidates. As TE dynamics have been observed to contribute to fungal genome plasticity and diversity, we calculated the proximity of candidate pathogenicity genes to the nearest TEs. 52.6% of these genes were flanked by or adjacent to TEs (i.e. at less than 5 kb distance; Supplementary Table 5). Comparison with genome-wide genes with regards to TE proximity revealed that candidate pathogenicity genes were significantly closer to TEs ($p < 0.05$; Fig. 4b). Gene diversification is also accelerated by meiotic recombination, which influences both natural population dynamics and selection. We used a combination of algorithms in two independent software (RDP4 and GARD) to detect signatures of recombination. We found signatures of recombination supported by both detection methods in a significant proportion (63.2%) of the

putative pathogenicity genes (Supplementary Table 6). Interestingly, genes with low or no signatures of recombination were in close proximity to TEs, whereas those with higher recombination events were found at greater distances from TEs (Fig. 4c). We hypothesize that recombination in particular and proximity to TEs promotes allelic diversity in genes that are putatively involved in quantitative pathogenicity.

## Pathogenicity genes are hotspots of positive diversifying selection

We assumed that candidate pathogenicity genes may have evolved under *R*-genes selective pressure on pathogen populations to escape recognition. Therefore, we searched for episodes of positive diversifying selection on protein-coding sequences by using the $\omega$ ($d_N/d_S$) parameter. Codon sites under positive selection are defined as those with elevated nonsynonymous/synonymous substitution ratios ($\omega > 1$) compared to the expectation under neutral evolution, $\omega = 1$. The proportion of positively-selected sites ($\omega > 1$) varied greatly among genes and ranged from 0.42% in the case of *Zt_1_01987* to 10.97% for *AvrStb6* (Supplementary Table 5). We also found four genes with a $d_N/d_S > 1$ (*Zt_3_00467*, $\omega = 7.03$; *AvrStb6*, $\omega = 2.84$; *Mycgr3G106276*, $\omega = 1.18$ and *AvrStb9*, $\omega = 1.04$), that could be considered as genes under strong selection pressure. Since $\omega$ averaged across all gene-tree branches rarely surpasses 1, and as the selective pressure could

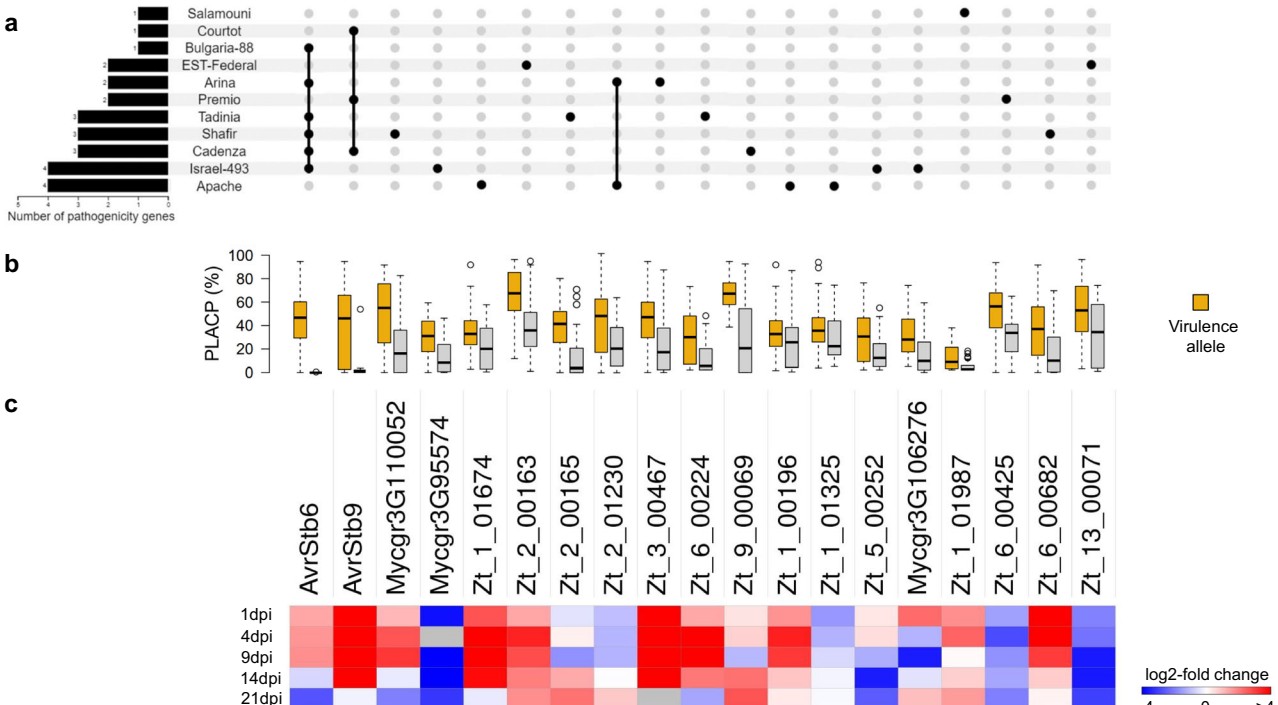

**Fig. 3 | The quantitative nature and host-specificity of Z. tritici pathogenicity.** **a** An upset plot of candidate pathogenicity genes across host cultivars. **b** Boxplots showing variation in quantitative pathogenicity (PLACP) of the isolates ($n = 103$ isolates) carrying different lead SNP alleles of the candidate pathogenicity genes (center line at the median, upper bound at 75th percentile, lower bound at 25th percentile, with whiskers chosen to show the 1.5 of the interquartile range). **c** Differential gene expression shown as log2-fold change compared to the culture medium. Red shades indicate up-regulated genes and blue shades indicate down-regulated genes during the infection cycle.

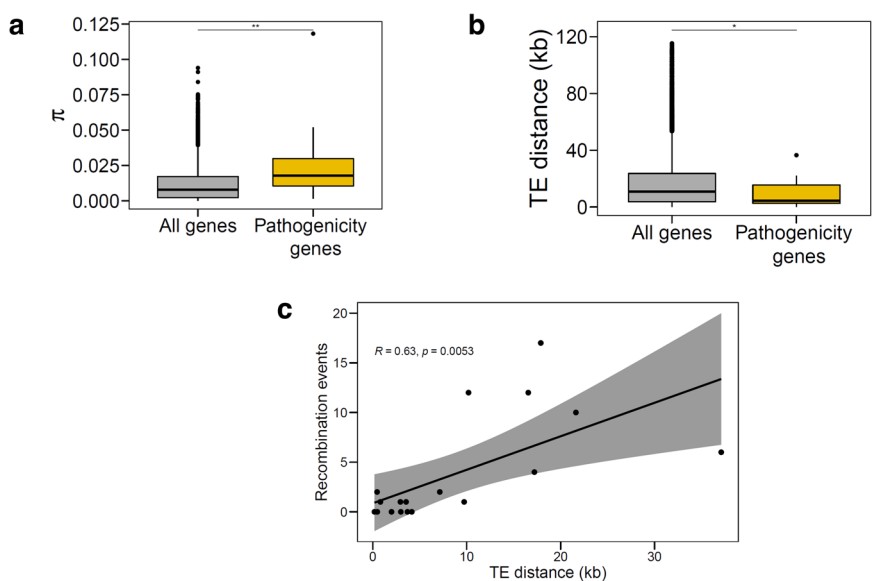

**Fig. 4 | Genetic diversity of quantitative pathogenicity genes in Z. tritici and factors contributing to gene diversification.** **a** Distribution of nucleotide diversity ($\pi$) among candidate pathogenicity genes ($n = 58$) and the rest of the protein-coding genes ($n = 10,993$). Statistical significance was determined using a permutation test with 1000 iterations; $p = 0.001818$**. **b** TE distance compared between pathogenicity genes ($n = 58$) and the rest of the protein-coding genes in Z. tritici ($n = 12,012$). Statistical significance was determined using a permutation test with 1000 iterations; $p = 0.04683$*. In (**a**) and (**b**) data are presented as box plots (center line at the median, upper bound at 75th percentile, lower bound at 25th percentile) with whiskers chosen to show the 1.5 of the interquartile range. **c** The relationship between TE distance and recombination events in candidate pathogenicity genes. The error band indicates the 95% confidence interval.

affect codon sites differently, we tested different codeML models to detect positively-selected codons. For all model comparisons, we found signatures of positive diversifying selection in 68.4% of candidate genes using both, the more conservative M1a vs. M2a and the M7 vs. M8 model pairs (Supplementary Table 5). Together, this shows that pathogenicity genes are hotspots of positive diversifying selection, likely as a result of a selection pressure exerted by host *R* genes.

## Two functionally distinct genes mediate quantitative pathogenicity

To determine the role of the candidate genes in quantitative pathogenicity, we selected two distinct genes for functional validation: *Zt_3_00467* (Fig. 5a), which encodes a SSP that is highly expressed *in planta*, with a peak at the disease transition phase (Fig. 5b); and *Zt_6_00682* (Fig. 6a), which encodes a S-adenosylmethionine-dependent methyltransferase (SAM-MTase) that is differentially expressed during the disease asymptomatic phase (Fig. 6b). For *Zt_3_00467*, we ectopically expressed the avirulent (IPO10273) and virulent (3D7) alleles in the 3D7 strain background and assessed quantitative pathogenicity traits on the host cultivars "Arina" and "Riband". Overall, Mutants expressing the avirulent allele ($3D7 + Zt\_3\_00467_{IPO10273}$) showed a slight reduction of PLACP levels compared to those expressing the virulent allele ($3D7 + Zt\_3\_00467_{3D7}$; Fig. 5c). When compared individually, one mutant expressing the avirulent allele showed a phenotype similar to the wild-type, while another expressing the virulent allele had higher PLACP compared to the wild-type (Supplementary Fig. 7). The difference in phenotypic responses between ectopic mutants is a well-known phenomenon that may be caused by distinct allele integration sites in the different mutant lines. Testing transformants expressing the avirulent allele on the cultivar "Riband" led to lower development of pycnidia compared to the virulent and the wild-type strains (Supplementary Fig. 8). "Arina" and "Riband" share a common resistance gene known as *Stb15*. These

phenotypic observations suggest that *Zt_3_00467* recognition leads to quantitative resistance which could be attributed to *Stb15*. Comparison of the avirulent and virulent protein sequences showed seven amino acid substitutions, all positively selected (Fig. 5d). Analysis of *Zt_3_00467* in the global *Z. tritici* collection showed that the gene was conserved and that most protein variants were population specific and segregating by geographical origin (Supplementary Fig. 9).

For *Zt_6_00682*, we generated knockout (KO) mutants of the 3D7 strain by targeted gene disruption and ectopic transformants with the virulent (3D7) and avirulent (3D1) alleles in the KO background. This allowed us to assess allele-dependent effects of *Zt_6_00682*. Three independent KO and ectopic transformants for each allele were tested on the cultivar "Shafir", where the association was detected. Compared to the wild-type, KO transformants resulted in unexpectedly high PLACP values ranging from 67% to 100% (Fig. 6c). Ectopic transformation of the 3D1 allele in the KO background ($3D7\Delta Zt\_6\_00682 + Zt\_6\_00682_{3D1}$) resulted in reduced levels of PLACP, while ectopic lines transformed with the 3D7 allele ($3D7\Delta Zt\_6\_00682 + Zt\_6\_00682_{3D7}$) showed a slightly higher PLACP than the wild-type (Fig. 6c). Besides "Shafir", the deletion mutants were aggressive on another four cultivars, but phenotypic differences between ectopic transformants and wild-type were not significant (Supplementary Fig. 6). Five amino acid differences were found between the avirulent and virulent protein sequences, of which three were under positive selection (Fig. 6d). Like *Zt_3_00467*, *Zt_6_00682*

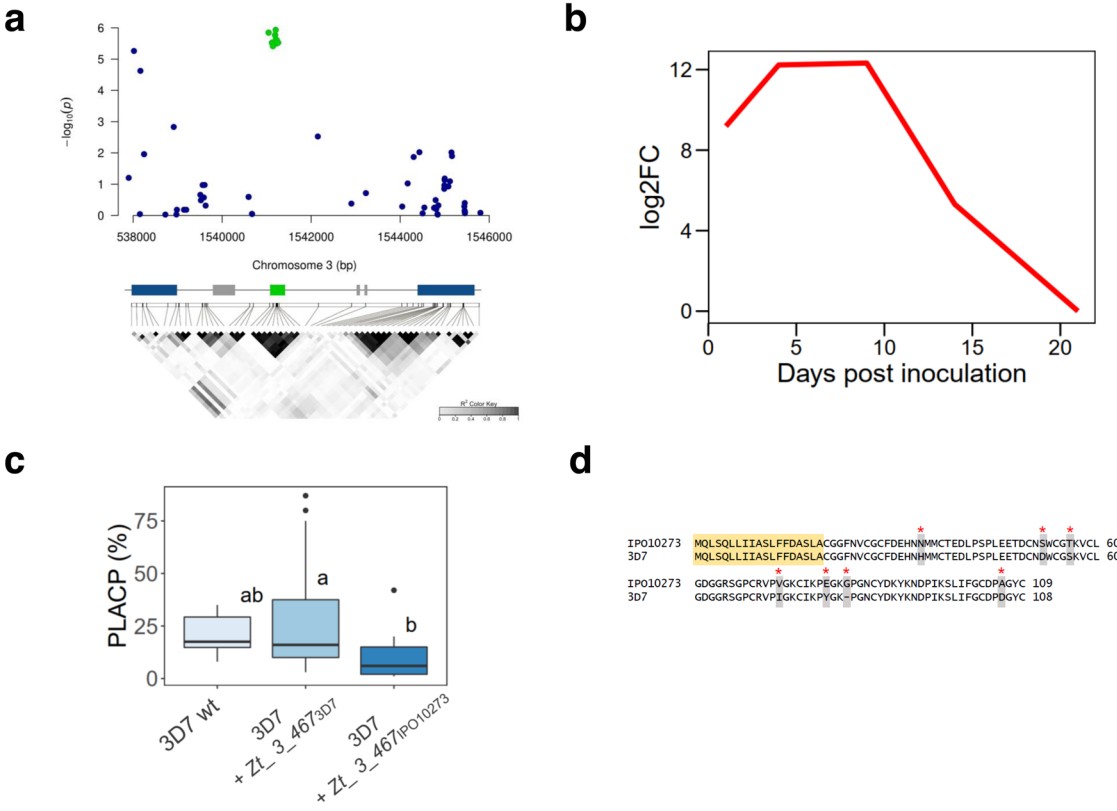

**Fig. 5 | An effector gene encoding a positively-selected small secreted protein is involved in quantitative pathogenicity of *Z. tritici*. a** Regional association plot of a candidate effector protein detected on cultivar "Arina". *X* and *Y* axes indicate positions on chromosome 3 and −log10 (*p* value) for associations with PLACP, respectively. Annotations on the reference genome are shown at the bottom with the candidate gene represented by a green box, other genes by blue boxes and transposable elements by gray boxes. A linkage disequilibrium plot is shown below the annotation bar. **b** log2-fold expression changes of *Zt_3_00467* based on RNAseq data collected throughout the time course of *Z. tritici* infection on wheat.
**c** Percentage of leaf area covered by pycnidia (PLACP) produced by the wild-type

strain 3D7 (*n* = 6 leaves) and the ectopic transformants expressing the virulent (3D7) and avirulent (IPO10273) alleles (*n* = 18 leaves each). Data are presented as box plots (center line at the median, upper bound at 75th percentile, lower bound at 25th percentile) with whiskers chosen to show the 1.5 of the interquartile range. Significant difference between group means are represented with different letters above the boxplots after a Tukey's HSD test. **d** Alignment of the virulent and avirulent protein sequences. Residues in yellow constitute the predicted signal peptide, gray residues show protein mutations and red asterisks show positively selected sites with a Bayes Empirical Bayes posterior probability >0.95.

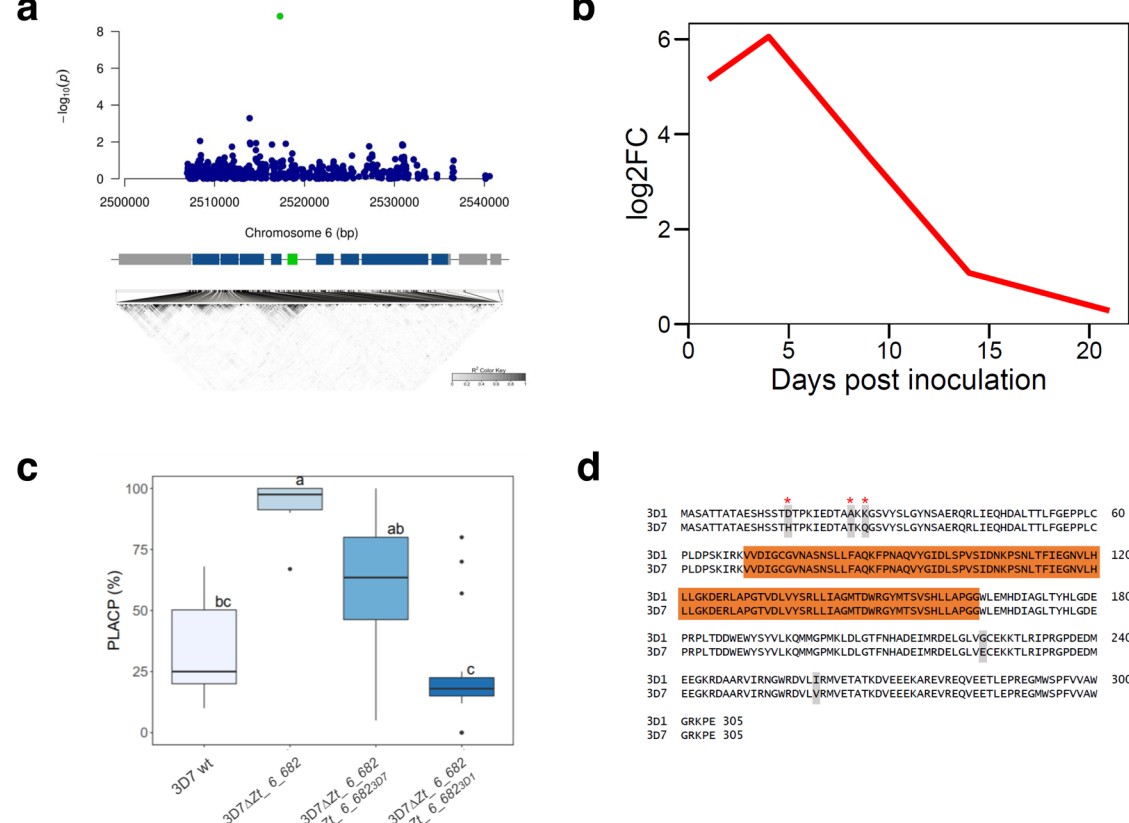

**Fig. 6 | A gene encoding a methyltransferase mediates quantitative pathogenicity in *Z. tritici*. a** Regional association plot of the pathogenicity gene detected on cultivar "Shafir". *X* and *Y* axes indicate positions on chromosome 6 and −log10 (*p* value) for associations with PLACP, respectively. Annotations on the reference genome are shown at the bottom with the candidate gene represented by a green box, other genes by blue boxes and transposable elements by gray boxes. A linkage disequilibrium plot is shown below the annotation bar. **b** log2-fold expression changes of *Zt_6_00682* based on RNAseq transcription data collected throughout the time course of *Z. tritici* infection on wheat. **c** Percentage of leaf area covered by pycnidia (PLACP) produced by the wild-type strain 3D7 (*n* = 6 leaves), knockout

mutants (*n* = 18 leaves) and the complementation mutants carrying the virulent (3D7) and avirulent (3D1) alleles (*n* = 18 leaves each). Significant difference between group means are represented with different letters above the boxplots after a Tukey's HSD test. Data are presented as box plots (center line at the median, upper bound at 75th percentile, lower bound at 25th percentile) with whiskers chosen to show the 1.5 of the interquartile range. **d** Alignment of the virulent and avirulent protein sequences. Residues in orange constitute the methyltransferase domain, gray residues show protein mutations and red asterisks show positively selected sites with a Bayes Empirical Bayes posterior probability >0.95.

was conserved in the global population but was highly polymorphic (Supplementary Fig. 9).

## Discussion

Our results point to a complex genetic architecture of pathogenicity in *Z. tritici*, involving many genes with variable additive effects, resulting in quantitative phenotypes on the host cultivar. These findings are consistent with previous studies[36,37] and provide further evidence that pathogenicity in this pathosystem is complex and results from the contribution of both large- and small-effect genes. We highlighted nineteen differentially and highly expressed genes *in planta* and focused on their features for subsequent investigations. Most of these genes had a peak expression in the phases corresponding to the colonization and the pre-pycnidia formation before the appearance of symptoms. During colonization, hyphae penetrate host tissue via stomata and colonize the substomatal cavity but remain exclusively in the intercellular space. This is followed by hyphal invasion of the apoplast leading to pre-pycnidia formation in substomatal cavities[26]. It was shown that these are the key phases of effector gene deployment by plant pathogens[38]. These generally encode small, secreted, cysteine-rich proteins with no sequence homology in other species and play a vital role in successful pathogen invasion, suppression of host immune perception and manipulation of its physiology[39]. Here, we mapped a

set of diverse secreted proteins. While effector-like proteins may initiate an effector-triggered immune response, as was shown for the gene *Zt_3_00467*, which triggered quantitative resistance in cultivars "Riband" and "Arina", secreted molecules with a known functional domain could also promote disease.

Of the functionally annotated secreted proteins, we identified domains associated with macromolecule degradation, fungal metabolism, or stress response, among others. Some genes encoded proteins that show strong homology with functionally characterized virulence factors from other pathogenic fungi. For instance, *Zt_1_01674* encodes a Thioredoxin reductase (*Trr*), a flavoenzyme that plays a central role in the thioredoxin oxidative stress resistance pathway and is involved in regulating gene transcription, cell growth and apoptosis[40]. Deletion of the *Trr* ortholog resulted in reduced hyphal growth, sexual reproduction, and virulence in *Fusarium graminearum*[41] and profoundly affected pathogenicity of the entomopathogenic fungus *Beauveria bassiana*[42]. We hypothesize that *Zt_1_01674* may play similar roles and could be an important factor in *Z. tritici* development and pathogenicity. We also identified a cell-wall degrading enzyme (*Zt_2_00163*) belonging to the Glycoside Hydrolase protein family. Glycoside hydrolases are known to play important roles in fungal invasion and nutrition acquisition via the degradation or modification of plant cell wall components, such as cellulose or hemicellulose[43].

Genes encoding proteins predicted to be non-secreted were common in our candidate list, highlighting not only the polygenic nature of pathogenicity, but also the potential role of the non-secretory pathway in fungal pathogenicity. This was demonstrated by the functional studies of *Zt_6_00682*, which encodes a SAM-dependent Methyltransferase (SAM-MTase). SAM-MTases catalyze the transfer of methyl groups from SAM to a large variety of substrates, ranging from metabolites to macromolecules, including proteins[44]. They are involved in epigenetic regulation with functions ranging from the control of gene expression, silencing or activation of transposon control[45]. *Zt_6_00682* shares some sequence similarities with the putative SAM-MTase gene *LaeA*, a component of the velvet transcription factor complex that functions as a global regulator of fungal secondary metabolism and development for a handful of fungi[46]. Recent work showed that mutants lacking the velvet gene *velB* in *Z. tritici* have defects in vegetative growth and asexual reproduction[47], while the loss of *LaeA* attenuates pathogenicity in a variety of filamentous fungi[48]. However, the inactivation of *Zt_6_00682* promoted pycnidia formation, suggesting that it functions differently from *LaeA* and may negatively regulate the transcriptional activity of genes or metabolites associated with host colonization. However, how this SAM-MTase affects the regulation of pathogenicity factors needs to be investigated as a key step toward understanding the evolutionary mechanisms associated with the emergence of highly aggressive strains.

In addition, we identified three genes with predicted transmembrane domains. For instance, the *DPM2* ortholog (*Zt_1_01325*) was highly expressed throughout the infection cycle, highlighting its potential role in *Z. tritici* pathogenicity. *DPM2* is a regulatory subunit of the dolichol-phosphate mannose synthase complex, key enzymes for GPI anchor biosynthesis[49]. The alteration of the *DPM* complex affected cell wall integrity, fungal growth and increased chitin levels in *Candida albicans*[49]. Traditionally, transmembrane proteins are systematically discarded in effector gene prediction studies. There is however growing evidence that they could play key roles in pathogenicity of filamentous fungi and our results support this hypothesis[50,51].

We found substantial evidence for host specificity demonstrated by a highly significant isolate × cultivar interaction and few shared associations between host cultivars. It is also probable that the weak overlap in candidate pathogenicity genes among host cultivars is partly due to a lack of statistical power to detect variants of small effect[52]. However, besides *Stb6*- and *Stb9*-carrying cultivars, there was no meaningful correlation among phenotypes on different hosts, which further supports host specificity. In several well studied host-pathogen systems, specificity is determined mainly by the matching of resistance or susceptibility genes and their corresponding effectors[53–55], following a GFG relationship (or inverse GFG in the case of necrotrophic fungal pathogens).

In *Z. tritici*, pathogenicity can be both qualitative[29,56,57] and quantitative[10,58]. Here, we showed that pathogenicity is predominantly quantitative based on multiple genes of small-to-moderate effects. This was demonstrated with the allele analysis of candidate genes and functional studies of *Zt_3_00467* and *Zt_6_00682*. In contrast to what has been shown for most GFG interactions, the recognition of the *Zt_3_00467* and *Zt_6_00682* avirulent alleles does not trigger strong resistance, but rather a reduction at the onset of disease symptoms. Furthermore, the fact that avirulent alleles are only recognized by certain wheat cultivars suggests that the interaction also follows a GFG model and demonstrates that *Zt_3_00467* and *Zt_6_00682* are likely quantitative avirulence factors interacting with specific host resistance genes. This is further supported by recent work in *Z. tritici* as shown for *Avr3D1* and a recently reported effector gene (*G_07189*) that triggers quantitative *Stb20q* resistance[11,59]. This is not uncommon in other filamentous fungi. For instance, the late effector LmSTEE98 from *Leptosphaeria maculans* triggers quantitative resistance following a GFG model[60], similarly, silencing three *Blumeria graminis* f sp. *tritici* effectors lead to quantitative gain of resistance[61]. Since *Zt_3_00467* encodes a typical effector protein and is recognized by hosts harboring the *Stb15* resistance gene, we suggest that *Zt_3_00467* could be *AvrStb15* and trigger an effector-induced defence. We hypothesize that *Stb15* resistance is not strong enough to block pathogen infection, although we cannot completely rule out the existence of a *Zt_3_00467* haplotype that triggers qualitative resistance. However, additional genetic factors might also contribute to differences in pathogenicity between avirulent and virulent strains. On the other hand, *Zt_6_00682* encodes an atypical avirulence factor and may interact indirectly with the host immune system. The fact that *Zt_6_00682* resembles genes involved in transcriptional regulation and that its deletion mutants are more aggressive on some cultivars, suggests that *Zt_6_00682* negatively regulates pathogenicity factors that are specifically recognized by one or more (unknown) host quantitative resistance gene(s). Complementation mutants further support host specificity, where recognition of the avirulent allele leads to a dramatic reduction in pycnidia formation. Unlike deletion mutants, this was only observed on the cultivar where the gene was detected initially ("Shafir"). However, the resistance gene involved and the underlying mechanisms through which the avirulent allele is recognized remain unknown.

In addition to *AvrStb6* and *AvrStb9*, we found the SSP *Zt_2_01230* to be significantly associated with variation in quantitative pathogenicity on two cultivars, suggesting a common yet unknown *Stb* resistance gene. We also identified a number of secreted and species-specific proteins. Since canonical fungal effectors are generally thought to be secreted, species-specific and without conserved protein domain[62], we propose that the ones identified here constitute a set of candidate effectors interacting with their cognate host *Stb* resistance products and should be a priority target for functional studies. Unlocking the molecular basis underpinning these interactions and how it could impact host defence would lead to a better exploitation of quantitative resistance, which is thought to be more durable[63].

Evolution and diversification of pathogenicity genes enables adaptation to new host cultivars and, in response to host population changes, they undergo changes in rapid allele frequency[64]. We observed high levels of genetic diversity in our candidate pathogenicity genes when compared to the genome-wide gene content. We found a single case of gene deletion and no significant impact of loss of function. Because of the intact and functional coding sequences for nearly all of the identified genes, we suggest that sequence polymorphism is the primary mechanism behind the rapid adaptation of *Z. tritici* to host populations, permitting the pathogen to circumvent host resistance while conserving the presumable function of the pathogenicity protein. However, it is important to note that the utilization of short-read sequencing technology imposes limitations on our ability to identify structural variants that may play a role in pathogenicity. Genes exhibiting presence/absence polymorphism could be underrepresented in our final set of candidate pathogenicity genes. We next investigated what could contribute to this extensive genetic diversity. Recombination is a central source of diversity in sexual fungal pathogens[65]. Here, we detected recombination events for a significant number of candidates, which suggests a role in the observed genetic diversity of pathogenicity genes. In *Z. tritici*, it was reported that recombination is a major driver of its evolution and is highly heterogeneous, concentrated in narrow genomic regions that are enriched in putative effector genes[8,66]. TEs are also thought to contribute to gene diversification by creating fast-evolving genomic regions through non-homologous recombination or repeat-induced point mutations[67]. Here, we found that candidate pathogenicity genes are indeed at closer distances to TEs, suggesting their potential role in sequence diversity. We also observed that TE distance was correlated with recombination events in putative pathogenicity genes, and that the levels of genetic diversity were high for genes with the highest

recombination events, regardless of their distances to TEs. We hypothesize that recombination is likely the major driver underlying the diversification of pathogenicity genes. A recent study supports this hypothesis where, unlike distance to TEs, the recombination rate was strongly associated with adaptive substitutions in *Z. tritici* genomes[4].

A key determinant of genetic diversity at pathogenicity genes is selection pressure. Understanding the evolution of pathogenicity in fungal populations with respect to the selection pressure imposed by host populations is necessary for the sustainable management of *R* genes in the field. This can be achieved by searching for signatures of positive diversifying selection at the risk of recovering some false positive signals. In this study, we used conservative maximum likelihood models and models that have been shown to be robust to recombination and significantly reduce false positives[68]. More than two thirds of candidate genes harbor signatures of positive selection acting on specific codons, suggesting that the diversifying selection is likely the primary evolutionary force maintaining non-synonymous substitutions relevant for *Z. tritici*-wheat quantitative interactions. For instance, the vast majority of different mutations between the avirulent and virulent versions of *Zt_3_00467* and *Zt_6_00682* have been identified as positively selected. Such selection pressure could be the result of host *R* gene recognition, and may have greatly contributed to the accumulation of beneficial mutations. This agrees with a previous study, which demonstrated that rapidly evolving genes play a prominent role in host-specific disease development[69]. Together, these results suggest that host–pathogen coevolution drives the emergence and maintenance of allelic diversity at loci likely involved in direct (or indirect) interactions with host *R* genes. Future surveys of the diversity and structure of *R* genes in wheat populations will provide a deeper insight into reciprocal patterns of selection, coevolution, and the maintenance of diversity in resistance and pathogenicity in natural populations.

## Methods

### Fungal material

We analyzed 103 *Z. tritici* isolates in this study deriving predominantly from a large collection of French isolates collected in 2009 and 2010. This genetically diverse sample was mainly isolated from the French cultivars "Apache" ($n = 44$) and "Premio" ($n = 42$). The remaining were isolated from cultivars "Soissons" ($n = 9$), "Caphorn" ($n = 3$), "Alixan" ($n = 1$), "Bermude" ($n = 1$), "Dinosor" ($n = 1$), and "Garcia" ($n = 1$). One isolate was collected in the United Kingdom in 2009 on cultivar "Humbert". All strains are monospore non-clonal isolates that were originally isolated either at the Plant Research International in the Netherlands or at INRAE UR BIOGER in France. For the functional validation of candidate pathogenicity genes, we used the two strains 3D1 (i.e., ST99CH3D1) and 3D7 (i.e., ST99CH3D7) collected in 1999 from a Swiss wheat field and previously used for the functional validation of *Avr3D1*[10].

### Plant material

The pathogenicity of the 103 isolates was initially evaluated on 29 wheat cultivars, including 15 carrying known *Stb* resistance genes. After the first replicate, only cultivars on which we observed phenotypic variability were kept for second and third replicates. These were the cultivars "Bulgaria-88" (*Stb1* and *Stb6*), "Israel-493" (*Stb3* and *Stb6*), "Tadinia" (*Stb4* and *Stb6*), "Shafir" and "Cadenza" (*Stb6*), "Estanzuela Federal" (*Stb7*), "Courtot" (*Stb9*), "Apache" (*Stb4/5* and *Stb11*), "Salamouni" (*Stb13*, *Stb14* and *StbSm3*) and "Arina" (*Stb6* and *Stb15*); the resistance background of cultivars "Premio" and "Toisondor" was not known. Details on the differential lines are available in Supplementary Table 3. This genetic material is available from the INRAE Small grain cereals Biological Resources Centre in Clermont-Ferrand (https://www6.clermont.inrae.fr/umr1095_eng/Organisation).

## Quantitative pathogenicity assays

For inoculum preparation, blastospores of *Z. tritici* stored at −80 °C were used to inoculate a liquid growth medium (Difco™ YPD Broth Cat. No. 242820) agitated at 140 rpm in a 17 °C climate cell for 144 h and transferred to agar growth medium (Difco™ YPD Agar Cat. No. 242720) for 96 h. Fresh blastospores were recovered with loops, resuspended in sterile water, and the inoculum was adjusted to a concentration of $10^6$ spores ml$^{-1}$, to which one drop of Tween®20 (Sigma-Aldrich® Cat. No. 9005-64-5) was added per 15 ml before inoculation. Five seeds from each cultivar were sown in 90*90*95 mm pots containing Floradur B soil (80% blond peat/20% black peat, with 1.2 kg m$^{-3}$ fertilizer 18/10/20). One day before inoculation, we kept three homogeneous seedlings on which a 75-mm-length portion was delimited on the first true leaf using a black marker. Sixteen days old seedlings were inoculated by applying the inoculum six times on the marked leaves with a flat tipped paintbrush (synthetic bristles, 12 mm width), depositing ~30,000 spores cm$^{-2}$ on the leaf surface. After inoculation, plants were watered and enclosed in transparent polyethylene high-density bags for 72 h to initiate the infection. Ten days post-inoculation, leaves were removed keeping only the marked leaf to ensure light homogeneity and longevity of the inoculated leaf. From sowing to disease scoring, plants were kept in growth chambers regulated at 80% relative humidity, 16 h of light per 24 h at a photon flux of 300 μmol m$^{-2}$ s$^{-1}$, 22 °C temperature under light and 18 °C in the darkness. Quantitative pathogenicity was assessed as percentages of leaf area covered by pycnidia (PLACP) and by necrosis (PLACN) at 21 days post inoculation, averaged over three leaves and three replicates (nine leaves per interaction). Phenotypic characterization of the generated deletion and ectopic transformants was done following the same procedure described above in two replicates.

## Whole-genome resequencing

Blastospores were harvested after 7 days growth in liquid cultures on yeast extract-peptone-dextrose Broth (Difco™ YPD Broth), and then lyophilized for 24 h. High molecular weight DNA was extracted using a phenol-chloroform-based procedure[70]. Illumina paired-end libraries were prepared for each sample and DNA fragments were re-sequenced with 2 × 100 bp reads on a HiSeq-2000 sequencing system at Genewiz, Inc. (formerly Beckman Coulter Genomics). Depth of sequence coverage was variable depending on the isolate, ranging from 16 to 303 genome equivalents with an average of 61 genome equivalents.

## SNP calling and quality filtering

For each isolate, reads were trimmed using trimmomatic v0.32[71] and then mapped to the reference genome IPO323[72] using the mem algorithm from BWA v0.7.7 with default settings[73]. Samtools v0.1.19 and Picard tools v1.106 were used to filter reads[74] (http://broadinstitute.github.io/picard/); to remove optical duplicates, secondary alignments, reads with a mapping quality below 30 and to keep only pairs in which both reads met quality checks. SNP calling was done with Freebayes v0.9 using the option --report-monomorphic[75] and all positions detected as low complexity regions or TEs were excluded. Low-complexity regions and TEs were detected using RepeatMasker with default settings (RepeatMasker Open-4.0 http://www.repeatmasker.org) and REPET v.2.5[76], respectively. Quality filters were applied to VCF files using in-house scripts. Depth of coverage (DP) calculated at each position followed a normal distribution around the depth main peak. Hence, a DP filter was set at twice the standard deviation of the mean. All VCFs were stored as a matrix reduced to keep only positions where at least one isolate carries an alternative allele at a non-filtered position. The SNP matrix was parsed as a.tped format, converted to.ped/.map. The final matrix was filtered for less than 50% missing data and more than 10% Minor Allele Frequency (MAF) using PLINK v.1.9[77] (https://cog-genomics.org/plink/).

## Linkage disequilibrium (LD)

To assess the mapping power and resolution of GWAS, we estimated genome-wide LD decay using PopLDdecay v.3.40[78]. The average squared Pearson's correlation coefficient ($r^2$) was computed between pairs of SNPs up to 20 kb and plotted over physical distance by fitting a nonlinear regression model.

## Population structure analyses

Two approaches were used to investigate the fungal population structure: (1) we performed a principal component analysis (PCA) using independent SNPs ($r^2 < 0.2$) with the --pca command PLINK v.1.9 and (2) the admixture model in STRUCTURE v.2.2 software[79] with a $10^5$ burn-in period, $5 \times 10^5$ MCMC steps and 5 iterations for each k value from 1 to 10. The optimal number of sub-populations was determined with the Evanno method[80].

## GWAS

For each quantitative trait, we conducted GWAS using single- and multi-locus linear mixed models based on the EMMA algorithm implemented in GAPIT v.3.0[81–83]. We used a VanRaden derived kinship matrix as a random effect, and set the PCA option to "true" to determine the optimum number of PCs to be included in the model, following the Bayesian Information Criterion (BIC). The most robust model was chosen by visualizing Quantile-Quantile plots that illustrate the observed vs. expected $p$ values. A uniform threshold of $p = 1/n$ ($n$ being the effective number of independent SNPs) was used as the genome-wide significance threshold. The effective number of independent SNPs was calculated using the Genetic type 1 Error Calculator; http://grass.cgs.hku.hk/gec/download.php. The suggestive genome-wide threshold for the 718,810 SNPs was $3.96 \times 10^{-6}$ and the significant $p$ value was $1.98 \times 10^{-7}$. We also adjusted SNP $p$ values to account for multiple testing correction using the false discovery rate (FDR) procedure. SNPs with an FDR-adjusted $p$ value < 0.10 were deemed significant. For LD analysis, we first calculated the pairwise LD with the lead SNP in a window size of 1 Mb using the --$r^2$ command in PLINK v.1.9. We set a cut-off of $r^2 > 0.5$ to determine candidate genomic regions and assigned predicted genes using a custom Perl script. In case more than one gene is assigned to the candidate region, we picked the closest one from the lead SNP having pathogenicity-like properties (e.g., secretion signal, differential expression *in planta*). If no gene fell in the LD region, we selected the closest gene to the lead SNP. LD heatmaps were constructed using Haploview v4.2[84].

## Functional annotation

We inferred gene models from the ab initio *Z. tritici* annotation[85] in the candidate genomic regions identified by GWAS. We used RNAseq datasets of the reference isolate IPO323[25] to check for missing genes or manually refine gene models (e.g., splice junctions, missing exons). N-terminal secretion signals were predicted using the Hidden Markov Model (HMM) scoring method implemented in SignalP 3.0[86]. Transmembrane domains were identified with TMHMM 2.0[87]. Effector probabilities were predicted through EffectorP 3.0[88]. Conserved protein domains were assigned using HMM scan against the PFAM 35.0 database and blastp was used to search for homologs from other fungi. Finally, we used PHIB-BLAST, a blast server for PHI-base (Pathogen-Host Interactions), to search for homologs against functionally characterized pathogenicity factors.

## Gene expression analysis

RNAseq data from in vitro and *in planta* (1-, 4-, 9-, 14- and 21-days post infection; dpi) were obtained from a previous study[25]. We computed gene expression quantified as fragments per kilobase of exon per million fragments mapped (FPKM) using the improved gene models with Cuffdiff v.2.2.1[89]. We inferred pairwise log2-fold expression changes and $p$ value adjustment for *in planta* RNAseq samples compared to the Czapek-Dox broth (CDB) culture medium. Genes with an FDR-adjusted $p$ value < 0.10 and a log2|fold change| > 2 were considered as differentially expressed during the infection.

## De novo assembly and sequence diversity

We used SPAdes v.3.14.1 to produce draft assemblies for each isolate[90]. We ran the tool with the following settings: -k 21,33,55,75,95 --careful to reduce mismatches and indel errors. We performed blastn[91] searches using ~1000 bp of the target locus as a query to retrieve matching contig segments in each assembly. High-confidence matches were first extracted from the contig set using Samtools v.1.10 and subsequently aligned with MAFFT v7.464 using the --auto option and 1000 iterative refinement cycles[92]. Alignments were inspected using JalView v.2.11.1[93] and ambiguous regions were removed with trimAL v1.2[94] with less stringent parameters. We used the batch mode of DNAsp v.6[95] to calculate summary statistics of population genetic parameters associated with candidate genes such as nucleotide diversities ($\pi$)[96] and Tajima's D[97]. TE locations for *Z. tritici* were retrieved from a previous study[85] and we parsed gene distances to TEs using the command "closest" from bedtools v.2.29.2[98].

## Evolutionary analysis

Before testing for positive diversifying selection, we checked for evidence of recombination, as this can cause false positives. We used the software packages RDP4.101[99] and GARD[100] for recombination detection. We considered genes with recombination events if both detection methods found evidence of recombination. We then constructed non-recombinant phylogenetic trees of gene coding sequences using the RAxML algorithm[101] implemented in RDP4. Statistical support for the nodes was obtained after 100 bootstrap replicates. We used codeml of the PAML v.4.9 package[102] to estimate the evolutionary rates of candidate genes, based on Markov models of codon evolution. Branch lengths of the inferred ML trees were estimated by using the model M0 and then fixed in further analyses. The model M0 assumes constant selective pressure across codon sites and over time. The selective pressure at the coding-sequence level was measured by $\omega$, the ratio of non-synonymous to synonymous rates ($d_N/d_S$). A $d_N/d_S$ ratio of 1 ($\omega = 1$) indicates neutrality, whereas $\omega < 1$ and $\omega > 1$ suggests purifying and diversifying selection, respectively. Models M1a, M2a, M7 and M8 of variable selective pressure across codon sites were used to estimate selective pressure and test for positive selection[103,104]. For each candidate gene, we performed pairwise likelihood ratio tests (LRTs) for positive selection, comparing models that allow sites with $\omega > 1$ (M2a and M8) with models that do not (M1a and M7). We considered genes under positive diversifying selection if both model comparisons in LRT were significant. Finally, positively selected sites were inferred with a Bayes Empirical Bayes (BEB) probability[104] and codons with $\omega > 1$ at a probability >0.95 were considered positively-selected.

## Functional validation of candidate pathogenicity genes

For *Zt_6_00682*, we generated knockout mutants in the 3D7 isolate and ectopic transformants expressing the virulent and avirulent alleles in the knock-out background. To create plasmid constructs for gene disruption, flanking regions (~1 kb before start and stop codons) of *Zt_6_00682* were amplified from gDNA using primers listed in Supplementary Table 7. The hygromycin resistance gene cassette was amplified from the plasmid pNOV2114_HygR_attP1_attP2 and cloned into a pNOV2214 vector. The vector was then digested with *Hind*III restriction enzyme (New England Biolabs), amplified, and used for the cloning of PCR fragments with a Gibson Assembly cloning kit (New England Biolabs) following the manufacturer's instructions. Constructs were transformed into *Escherichia coli* NEB 5-α using a heat shock transformation, screened by PCR and verified by Sanger sequencing using primers listed in Supplementary Table 7. Confirmed plasmids were transformed into *Agrobacterium tumefaciens* cells by heat shock

and validated by PCR. The vector was used to transform *Z. tritici* via *A. tumefaciens*-mediated transformation (ATMT) following a procedure adapted from Zwiers and de Waard[105]. Transformed *Z. tritici* colonies growing in the presence of hygromycin were tested by PCR to confirm the successful deletion of the gene. To validate the allele-dependent effect of *Zt_6_00682*, we generated and validated ectopic transformants by placing both the virulent (3D7) and the avirulent (3D1) alleles into the 3D7 KO background, following the same procedure described above with the following modifications: (1) we used the vector pNOV_3Gate_SUL with sulfonylurea as a resistance marker and the digestion was achieved with *Hind*III and *Kpn*I restriction enzymes. The resulting vectors, pNOV_sulf_3D7g.7232 and pNOV_sulf_3D1g.7155, were used to transform *Z. tritici* via ATMT following the procedure described above. Primers used to verify gene disruptions and ectopic integrations are listed in Supplementary Table 7. Finally, three independent transformants carrying the different virulence alleles and three knock-out mutants were used for phenotypic characterization *in planta*. For *Zt_3_00467*, we ectopically expressed the virulent (3D7) and avirulent (IPO10273) alleles in the 3D7 background. A fragment containing the *Zt_3_00467* gene, including 1800 bp upstream of the start codon and 463 bp downstream of the stop codon was amplified using Phusion DNA polymerase (NEB) and the primers included in Supplementary Table 7. This fragment was cloned into a pCGEN vector[106] previously digested with *Kpn*I (New England Biolabs), using the In-Fusion HD Cloning Kit (Takara Bio) and following the manufacturer's instructions. Vectors were transformed using heat-shocked *E. coli* Stellar cells and verified using Sanger sequencing. Transformation via ATMT was performed as described above and selection for the *Z. tritici* mutants was performed on geneticin (150 μg ml$^{-1}$)-containing plates.

### Reporting summary

Further information on research design is available in the Nature Portfolio Reporting Summary linked to this article.

## Data availability

The sequence and annotation of *Zymoseptoria tritici* IPO323 reference genome are available from DOE Joint Genome Institute website: https://mycocosm.jgi.doe.gov/Zymtr1/Zymtr1.home.html. The sequence data generated in this study have been deposited in the NCBI Sequence Read Archive under the accession number PRJNA777581. Sequence data for the global isolates are available at the NCBI Sequence Read Archive under the accession number PRJNA327615. The processed summary GWAS and gene expression data are provided in Supplementary Data files 1 and 2.

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

## Acknowledgements

This work was supported by the French National Research Agency (program BIOADAPT, ANR-12-ADAP-0009-04) for T.C.M. and J.A., the FSOV 2008 B project for T.C.M., and BASF France SAS for R.A. post-doctoral fellowship. T.C.M. was supported by a Short-Term Scientific Mission granted by the COST SUSTAIN action FA1208. A.S.V. was supported by the Ministerio de Ciencia e Innovación (Ramón y Cajal program, grant no. RYC2018-025530-I). INRAE BIOGER benefits from the support of Saclay Plant Sciences-SPS (ANR-17-EUR-0007). We thank David Gouache (ARVALIS Institut du Végétal, Boigneville, France), Marc-Henri Lebrun, Henriette Goyeau, Frédéric Suffert and Anne-Sophie Walker (INRAE UR BIOGER, Palaiseau, France) for their help in establishing the fungal collection; Robert King and Jason Rudd (Rothamsted Research, Harpenden, United Kingdom) for providing RNAseq data; Gabriel Scalliet (Syngenta Crop Protection AG, Stein, Switzerland) for providing the vectors pNOV2114, pNOV_3Gate_SUL and

pNOV_3Gate_HYG used to clone the candidate pathogenicity genes; Martin Willigsecker and Béatrice Beauzoone (INRAE UR BIOGER) for their help in the pathology assays; and Thierry Rouxel (INRAE UR BIOGER) for critical reading of the manuscript and valuable feedback.

## Author contributions

T.C.M. and R.A. conceived and designed the study. G.H.J.K. and H.G. isolated and provided most of the natural *Z. tritici* isolates constituting the GWAS panel. A.D., T.C.M. and Jo.C. carried out pathogenicity assays on the natural isolates, and R.A. on the transformant isolates. A.D. and T.C.M. prepared the DNA samples for whole-genome resequencing. Je.C., N.L. and J.A. performed the bioinformatics analyses, including SNP calling and de novo genome assemblies. D.C. provided sequence data of the global *Z. tritici* collection. R.A., L.D., A.P. and A.S.V. performed the cloning and genetic transformations. R.A. performed the genome-wide association, gene expression and evolutionary sequence analyses. R.A. and T.C.M. wrote the manuscript. All authors have reviewed the manuscript.

## Competing interests

The authors declare no competing interests.
