## [Peer Review File · Nature Communications]

REVIEWER COMMENTS

Reviewer #1 (Remarks to the Author):

In this study, the authors utilize GWA to investigate the ability of Zymoseptoria to attack multiple wheat genotypes. Using these candidate genes, the authors conduct a series of population genetic tests and work to preliminarily validate a couple of the candidates. This is a very interesting study but I have a couple of concerns about over-interpretation of SNP causality and how to test population genetic terms that may influence the observations.

For the DNA sequencing, do the authors have an estimate of how reliable the illumina sequencing is at finding the presence absence of whole genes (Rather than small indels). There is work being released in multiple systems showing that short-read sequencing can struggle with structural variants and other aspects of the pangenome. These can often be better GWA hits and some information to help guide the readers interpretation of the genomes would be helpful. For example how confident should we be that the TE position in the Grandaubert 2015 paper are accurate for a diverse collection of accessions? Work in other Fungi shows that TE copy/position is changing quite rapidly and can be associated with host preference.

In lines 46-49, has there been a fully unbiased quantification of where the adaption is occurring to claim that it is mainly at effectors? There is also a large collection of genes being found that modulate the pathogens ability to resist plant compounds that involve genes that are not effectors. I ask because most techniques are biased towards TypeIII secreted proteins or genes with a specific effector domain. As such, it isn't quite clear if it is possible to claim where most of adaptation/evolution is occurring in a truly unbiased format if the genes studied are studied based on a limited a priori information set. This topic is somewhat broached in the second paragraph (starting line 63) but never really connected in an explicit way to the dominant effector concept in the first paragraph.

Line 75, did effector discovery lead to quantitative resistance gene identifications? There are a number of quantitative resistance studies using GWA in plant pathogens that would seem to inform this section however the only studies listed are for Zymoseptoria which seems odd. I understand that there is a large group of researchers studying this pathogen but it is not the only one being used to try and understand quantitative aspects of fungal virulence. It would seem that admitting that some of this work is occurring would help to connect the very broad introduction to the sole focus on Zymoseptoria at the end of the introduction. Especially as some of this work also focuses on how the hosts variation influences the results and involves transcriptomics on the pathogen. This is reinforced in the discussion where there is no mention of other work on quantitative virulence studies in other fungal/GWA systems.

Line 136-137, while LD gives an indication of recombination, there are still a large number of phenomena (heterogeneity, epistasis, trans-ld, large SNP/Low individual problem, etc.) that can lead to a lost of gene level resolution within GWA studies. I'm not sure given the recent progress in understanding GWA that it is safe to presume that GWA will give the causal gene without additional support.

Along the same lines, It is likely not safe to presume that the SNP is causal as is being done in line 225-226 where it is argued that most of the variation is likely sequence changes. Especially given the local LD blocks shown in Figure 5 and 6, if these are characteristic of the overall typical positive region, then it appears that haplotypes are being identified and that it isn't clear what is actually the causal polymorphism. As such, I would suggest more caution in making claims about the type of polymorphism that is underlying causality.

Line 188 – The section on the transcriptome is almost too tightly written. I reread this section several times and was unsure about how the genes at the start (GWA candidates?) end up being represented at the end of this same section. This might be because the section uses candidate rather than GWA candidate vs transcriptome candidate so I wasn't always following where the support for the gene was coming. A bit of rewording to clarify this section would help a reader who is not an expert.

Line 223 – I'm not sure if the null set for this test should be all genes in the dataset. For example, gene without any SNPs in the dataset could never be a GWA candidate and as such would not come into the pathogenicity dataset. They would however alter the π estimate for all genes if included. Given the noisiness in gene sets, the standard for this type of test is a permutation test where a similar set of genes is pulled at random from the whole genome to compare how often a similar distribution of variances is identified. Given long-tailed skewed distributions in gene sequence variation estimates within a genome it is often quite easy to pull out a set of genes that gives a significant Mann-Whitney test. Honestly, I'm not sure if the significance will or will not change given the distribution of π values in the whole genome. But given that the virulence genes are not in the extreme tail of the distributions, a permutation test is more appropriate.

Line 234-238, I'm a bit confused with this section on recombination. The start says that LD rapidly decays which would be via recombination. Yet this section says that there is very little recombination in the genes that are being found. Are the genes being found in regions of extended LD in comparison to the rest of the genome? As such, it is possible that they are in regions of selective sweeps which could confuse the GWA. Alternatively, the recombination algorithms are known to struggle with estimating recombination and it would likely be useful to do a visual comparison of the loci to ensure that the lack of recombination is real.

If there is a lack of recombination, then the π estimates start getting a bit messy as that would suggest that there is a greater separation of haplotypes at these loci than the average locus as recombination can alter variance estimates at different ranges (e.g. <https://www.ncbi.nlm.nih.gov/pmc/articles/PMC6906284/>).

Line 261-263, is the signal for positive selection only in the signal peptides? I ask because signal peptides don't have a specific sequence but instead a general physical property on which their function relies. As such, the synonymous/nonsynonymous ratio in these is typically greater than for a typical protein (i.e. TF or enzyme) where the specific sequence and structure have a tighter relationship. I'm also a bit puzzled as to why the signal peptide would be under diversifying selection? Or am I misreading and it is the whole protein that is showing this signal?

Minor comments

Line 172, I'm having a bit of trouble with the average phenotypic variance being >9% in all cases given the number of loci found per trait. Is the total variance adding up to >100% for a trait. This might arise from the way the section starts in line 155 where all the traits GWA are described across all traits but it isn't quite clear in the text how many traits this is actually representing and how many SNPs are found per trait. This is somewhat shown in Figure 2 but the text could be helped by representing this.

Line 173, is $R^2 < 30\%$ or absolute 30?

Reviewer #2 (Remarks to the Author):

The goal of this manuscript was to use genomic sequencing of 103 isolates of *Zymoseptoria tritici* in a GWAS analysis to identify quantitative pathogenicity. 58 candidate genes were identified and expression analysis was used to identify those with likely effects. Most showed signatures of diversifying selection and two were cloned for functional analysis. A conclusion was that many of the QTL likely showed specific interactions with host cultivars and had differing modes of action.

The manuscript overall was very well done and the conclusions were sound. It could be improved by adding hypothesis testing to the last paragraph of the Introduction so that the reader can know exactly which knowledge gaps are being addressed and how the results advance our understanding. Lines 111-118 provided a nice summary of what was done, but it would be much better to provide one or more specific, scientific hypotheses that can be tested with the data. This will be easy to add as there are several obvious possibilities. Testing of clear hypotheses will provide greater clarity about exactly what you have done and how it moves the science forward.

In lines 402-406 the authors might want to look up *Front. Microbiol.* 2019; 10: 2210, in which the velvet gene was knocked out in *Z. tritici*. The References have some variability in formatting – scientific names not italicized, all words capitalized – that should be corrected. I circled those on the first two pages but the authors can check after that. To save time typing, I wrote out most of my comments – which in general were very minor – on a hard copy of the manuscript and will return them in a scan. Hope you can read my writing. Very nice job overall.

REVIEWER COMMENTS

Reviewer #1 (Remarks to the Author):

In this study, the authors utilize GWA to investigate the ability of *Zymoseptoria* to attack multiple wheat genotypes. Using these candidate genes, the authors conduct a series of population genetic tests and work to preliminarily validate a couple of the candidates. This is a very interesting study but I have a couple of concerns about over-interpretation of SNP causality and how to test population genetic terms that may influence the observations.

For the DNA sequencing, do the authors have an estimate of how reliable the illumina sequencing is at finding the presence absence of whole genes (Rather than small indels). There is work being released in multiple systems showing that short-read sequencing can struggle with structural variants and other aspects of the pangenome. These can often be better GWA hits and some information to help guide the reader's interpretation of the genomes would be helpful. For example how confident should we be that the TE position in the Grandaubert 2015 paper is accurate for a diverse collection of accessions? Work in other Fungi shows that TE copy/position is changing quite rapidly and can be associated with host preference.

In lines 46-49, has there been a fully unbiased quantification of where the adaptation is occurring to claim that it is mainly at effectors? There is also a large collection of genes being found that modulate the pathogens ability to resist plant compounds that involve genes that are not effectors. I ask because most techniques are biased towards TypeIII secreted proteins or genes with a specific effector domain. As such, it isn't quite clear if it is possible to claim where most of adaptation/evolution is occurring in a truly unbiased format if the genes studied are studied based on a limited a priori information set. This topic is somewhat broached in the second paragraph (starting line 63) but never really connected in an explicit way to the dominant effector concept in the first paragraph.

Line 75, did effector discovery lead to quantitative resistance gene identifications? There are a number of quantitative resistance studies using GWA in plant pathogens that would seem to inform this section however the only studies listed are for *Zymoseptoria* which seems odd. I understand that there is a large group of researchers studying this pathogen but it is not the only one being used to try and understand quantitative aspects of fungal virulence. It would seem that admitting that some of this work is occurring would help to connect the very broad introduction to the sole focus on *Zymoseptoria* at the end of the introduction. Especially as some of this work also focuses on how the hosts variation influences the results and involves transcriptomics on the pathogen. This is reinforced in the discussion where there is no mention of other work on quantitative virulence studies in other fungal/GWA systems.

Line 136-137, while LD gives an indication of recombination, there are still a large number of phenomena (heterogeneity, epistasis, trans-ld, large SNP/Low individual problem, etc.) that can lead to a loss of gene level resolution within GWA studies. I'm not sure given the recent progress in understanding GWA that it is safe to presume that GWA will give the causal gene without additional support.

Along the same lines, It is likely not safe to presume that the SNP is causal as is being done in line 225-226 where it is argued that most of the variation is likely sequence changes. Especially given the local LD blocks shown in Figure 5 and 6, if these are characteristic of the overall typical positive region, then it appears that haplotypes are being identified and that it isn't clear what actually the causal polymorphism is. As such, I would suggest more caution in making claims about the type of polymorphism that is underlying causality.

Line 188 – The section on the transcriptome is almost too tightly written. I reread this section several times and was unsure about how the genes at the start (GWA candidates?) end up being represented at the end of this same section. This might be because the section uses candidate rather than GWA candidate vs transcriptome candidate so I wasn't always following where the support for the gene was coming. A bit of rewording to clarify this section would help a reader who is not an expert.

Line 223 – I'm not sure if the null set for this test should be all genes in the dataset. For example, gene without any SNPs in the dataset could never be a GWA candidate and as such would not come into the pathogenicity dataset. They would however alter the π estimate for all genes if included. Given the noisiness in gene sets, the standard for this type of test is a permutation test where a similar set of genes is pulled at random from the whole genome to compare how often a similar distribution of variances is identified. Given long-tailed skewed distributions in gene sequence variation estimates within a genome it is often quite easy to pull out a set of genes that gives a significant Mann-Whitney test. Honestly, I'm not sure if the significance will or will not change given the distribution of π values in the whole genome. But given that the virulence genes are not in the extreme tail of the distributions, a permutation test is more appropriate.

Line 234-238, I'm a bit confused with this section on recombination. The start says that LD rapidly decays which would be via recombination. Yet this section says that there is very little recombination in the genes that are being found. Are the genes being found in regions of extended LD in comparison to the rest of the genome? As such, it is possible that they are in regions of selective sweeps which could confuse the GWA. Alternatively, the recombination algorithms are known to struggle with estimating recombination and it would likely be useful to do a visual comparison of the loci to ensure that the lack of recombination is real.

If there is a lack of recombination, then the π estimates start getting a bit messy as that would suggest that there is a greater separation of haplotypes at these loci than the average locus as recombination can alter variance estimates at different ranges (e.g. <https://www.ncbi.nlm.nih.gov/pmc/articles/PMC6906284/>).

Line 261-263, is the signal for positive selection only in the signal peptides? I ask because signal peptides don't have a specific sequence but instead a general physical property on which their function relies. As such, the synonymous/nonsynonymous ratio in these is typically greater than for a typical protein (i.e. TF or enzyme) where the specific sequence and structure have a tighter relationship. I'm also a bit puzzled as to why the signal peptide would be under diversifying selection? Or am I misreading and it is the whole protein that is showing this signal?

Minor comments:

Line 172, I'm having a bit of trouble with the average phenotypic variance being >9% in all cases given the number of loci found per trait. Is the total variance adding up to >100% for a trait. This might arise from the way the section starts in line 155 where all the traits GWA are described across all traits but it isn't quite clear in the text how many traits this is actually representing and how many SNPs are found per trait. This is somewhat shown in Figure 2 but the text could be helped by representing this.

Line 173, is R^2 <30% or absolute 30?

Reviewer #2 (Remarks to the Author):

The goal of this manuscript was to use genomic sequencing of 103 isolates of *Zymoseptoria tritici* in a GWAS analysis to identify quantitative pathogenicity. 58 candidate genes were identified and expression

analysis was used to identify those with likely effects. Most showed signatures of diversifying selection and two were cloned for functional analysis. A conclusion was that many of the QTL likely showed specific interactions with host cultivars and had differing modes of action.

The manuscript overall was very well done and the conclusions were sound. It could be improved by adding hypothesis testing to the last paragraph of the Introduction so that the reader can know exactly which knowledge gaps are being addressed and how the results advance our understanding. Lines 111-118 provided a nice summary of what was done, but it would be much better to provide one or more specific, scientific hypotheses that can be tested with the data. This will be easy to add as there are several obvious possibilities. Testing of clear hypotheses will provide greater clarity about exactly what you have done and how it moves the science forward.

In lines 402-406 the authors might want to look up *Front. Microbiol.* 2019; 10: 2210, in which the velvet gene was knocked out in *Z. tritici*. The References have some variability in formatting – scientific names not italicized, all words capitalized – that should be corrected. I circled those on the first two pages but the authors can check after that. To save time typing, I wrote out most of my comments – which in general were very minor – on a hard copy of the manuscript and will return them in a scan. Hope you can read my writing. Very nice job overall.

Responses to reviewers:

Reviewer #1 (Remarks to the Author):

COMMENT: *In this study, the authors utilize GWA to investigate the ability of Zymoseptoria to attack multiple wheat genotypes. Using these candidate genes, the authors conduct a series of population genetic tests and work to preliminarily validate a couple of the candidates. This is a very interesting study but I have a couple of concerns about over-interpretation of SNP causality and how to test population genetic terms that may influence the observations.*

For the DNA sequencing, do the authors have an estimate of how reliable the illumina sequencing is at finding the presence absence of whole genes (Rather than small indels). There is work being released in multiple systems showing that short-read sequencing can struggle with structural variants and other aspects of the pangenome. These can often be better GWA hits and some information to help guide the reader's interpretation of the genomes would be helpful. For example how confident should we be that the TE position in the Grandaubert 2015 paper is accurate for a diverse collection of accessions? Work in other Fungi shows that TE copy/position is changing quite rapidly and can be associated with host preference.

RESPONSE: Thank you very much for this feedback. We appreciate the concerns raised by the reviewer and we fully agree that short-reads can struggle with structural variant detection and other genome assembly aspects; however, these were not our intentions in this study. We do not have an exact estimate on the reliability of Illumina short-reads in finding gene presence/absence, but this sequencing technology was shown to be robust (low error rates) in yielding high-confidence SNPs and is sufficient enough to build good quality de novo genome assemblies for gene sequence extraction, in case of filamentous fungi with relatively small genomes (as shown in diverse research work in *Z. tritici* and various other fungal organisms). The sequence data generated here has an average depth of 61x and we only kept reads with a minimum depth of 30x across the genome, which is sufficiently reliable for downstream analysis (rule of thumb being 30x). We have also applied robust algorithms (for variant calling and genome assemblies) and bioinformatics/statistical tools to process sequence data, and hard quality filters to extract high-confidence

SNPs and genes sequence. For instance, we only used filtered SNPs in our analyses, but no indels (there is a significant discrepancy in indel calling algorithms, and they could be prone to false positives). For gene sequences, we were actually successful in extracting all of the candidate sequences from draft assemblies, with one exception (Mycgr3G110052 gene, which was affected by presence/absence polymorphism). Isolates showing truncated sequence, ambiguous or misassembled sequences were excluded. The quality of our gene sequence extraction was also confirmed with Sanger sequencing of the isolates used for functional studies of two separate genes. With regards to TEs positions, we fully agree with the statement made above about TE copy/position. However, our intentions here were to provide an approximate picture of the genomic environment where candidate pathogenicity genes reside using the reference genome, rather than the exact distances to TEs (using individual genomes). This could prove hard to get, especially with short-reads, given the complexity in assembling those regions. However, previous studies on *Z. tritici* and similar organisms show that genes residing in TE-rich regions are generally found in a similar genomic environment in different isolates. We hope that these explanations address the reviewer's concerns.

COMMENT: *In lines 46-49, has there been a fully unbiased quantification of where the adaptation is occurring to claim that it is mainly at effectors? There is also a large collection of genes being found that modulate the pathogens ability to resist plant compounds that involve genes that are not effectors. I ask because most techniques are biased towards Type III secreted proteins or genes with a specific effector domain. As such, it isn't quite clear if it is possible to claim where most of adaptation/evolution is occurring in a truly unbiased format if the genes studied are studied based on a limited a priori information set. This topic is somewhat broached in the second paragraph (starting line 63) but never really connected in an explicit way to the dominant effector concept in the first paragraph.*

RESPONSE: Indeed, and we completely agree with this statement. In fact, it is one of the main points raised in this paper; and added evidence to this by showing that a large number of genes detected in our analyses involve genes that are not necessarily secreted effectors. This was the case for example of the methyltransferase gene that we functionally validated. However, to avoid the confusion or bias in this paragraph, we have rephrased the sentence by replacing the words 'effector loci' to 'pathogenicity genes'.

COMMENT: *Line 75, did effector discovery lead to quantitative resistance gene identifications? There are a number of quantitative resistance studies using GWA in plant pathogens that would seem to inform this section however the only studies listed are for Zymoseptoria which seems odd. I understand that there is a large group of researchers studying this pathogen but it is not the only one being used to try and understand quantitative aspects of fungal virulence. It would seem that admitting that some of this work is occurring would help to connect the very broad introduction to the sole focus on Zymoseptoria at the end of the introduction. Especially as some of this work also focuses on how the hosts variation influences the results and involves transcriptomics on the pathogen. This is reinforced in the discussion where there is no mention of other work on quantitative virulence studies in other fungal/GWA systems.*

RESPONSE: Thank you for raising this point. Yes, there are few examples of discovered effectors leading to quantitative plant resistance, and we have now provided examples other than *Z. tritici* (e.g., *Leptosphaeria maculans*, *Blumeria graminis*) in the discussion as requested (see lines 472-477). We totally agree with the fact that there are a number of studies dealing with quantitative resistance using GWA in plant pathogens as we mentioned in lines 86-87; however, this is not the case of quantitative virulence in fungal plant pathogens which remains underexplored. That is why we preferred to introduce broadly the concepts of quantitative resistance/quantitative virulence and highlight the knowledge gap between the two in this section of the introduction, without a special focus on a given pathosystem. We hope that this clarifies our intentions, and responds to this comment.

COMMENT: *Line 136-137, while LD gives an indication of recombination, there are still a large number of phenomena (heterogeneity, epistasis, trans-ld, large SNP/Low individual problem, etc.) that can lead to a loss of gene level resolution within GWA studies. I'm not sure given the recent progress in understanding GWA that it is safe to presume that GWA will give the causal gene without additional support. Along the same lines, It is likely not safe to presume that the SNP is causal as is being done in line 225-226 where it is argued that most of the variation is likely sequence changes. Especially given the local LD blocks shown in Figure 5 and 6, if these are characteristic of the overall typical positive region, then it appears that haplotypes are being identified and that it isn't clear what actually the causal polymorphism is. As such, I would suggest more caution in making claims about the type of polymorphism that is underlying causality.*

RESPONSE: Thank you for pointing this out. We totally agree with the statement that GWA alone is not sufficient to give the causal gene and that other phenomena may impact causal gene identification. Here, we simply wanted to state that our marker coverage exceeds genome-wide LD decay (a critical factor affecting GWAS hits) and that our high SNP density vs. average gene distance, would allow associations to be identified at the gene level, but certainly, with no additional clues about the causality. With this in mind, we were very careful in interpreting the candidate genes identified in this study (even in the validated genes, since multiple SNPs are involved and we do not know which is/are responsible for trait variations), and we tried to provide multiple lines of evidence (e.g. transcriptomics, functional data) to support their potential role in quantitative pathogenicity variations.

In fact, we have not used the term "causal" in the manuscript, with the exception of lines 104-106, where we provide the general aim of GWA studies. For extra caution, we have revised the manuscript to make sure there is no over-interpretation of SNP causality and removed the lines 225-226 from the text.

COMMENT: *Line 188 – The section on the transcriptome is almost too tightly written. I reread this section several times and was unsure about how the genes at the start (GWA candidates?) end up being represented at the end of this same section. This might be because the section uses candidate rather than GWA candidate vs transcriptome candidate so I wasn't always following where the support for the gene was coming. A bit of rewording to clarify this section would help a reader who is not an expert.*

RESPONSE: Thank you for raising this point. We have now reworded this section on gene expression to facilitate the reading and understanding.

COMMENT: *Line 223 – I'm not sure if the null set for this test should be all genes in the dataset. For example, gene without any SNPs in the dataset could never be a GWA candidate and as such would not come into the pathogenicity dataset. They would however alter the P_i estimate for all genes if included. Given the noisiness in gene sets, the standard for this type of test is a permutation test where a similar set of genes is pulled at random from the whole genome to compare how often a similar distribution of variances is identified. Given long-tailed skewed distributions in gene sequence variation estimates within a genome it is often quite easy to pull out a set of genes that gives a significant Mann-Whitney test. Honestly, I'm not sure if the significance will or will not change given the distribution of P_i values in the whole genome. But given that the virulence genes are not in the extreme tail of the distributions, a permutation test is more appropriate.*

RESPONSE: This is a very interesting suggestion that we totally agree with. We have now performed permutation tests with 1000 iterations, and updated the results and methods accordingly. However, this did not change the statistical significance of group comparisons.

COMMENT: *Line 234-238, I'm a bit confused with this section on recombination. The start says that LD rapidly decays which would be via recombination. Yet this section says that there is very little recombination*

in the genes that are being found. Are the genes being found in regions of extended LD in comparison to the rest of the genome? As such, it is possible that they are in regions of selective sweeps which could confuse the GWA. Alternatively, the recombination algorithms are known to struggle with estimating recombination and it would likely be useful to do a visual comparison of the loci to ensure that the lack of recombination is real. If there is a lack of recombination, then the P_i estimates start getting a bit messy as that would suggest that there is a greater separation of haplotypes at these loci than the average locus as recombination can alter variance estimates at different ranges (e.g. <https://www.ncbi.nlm.nih.gov/pmc/articles/PMC6906284/>).

RESPONSE: We apologize for this confusion. What we actually report here, is that we found signatures of recombination in a significant number of GWAS genes, in agreement with the LD analysis (genome-wide and around lead SNPs; data not shown with the exception of Fig. 5 and Fig. 6). The main goal of this section is to provide clues about the drivers of the observed high genetic diversity in putative pathogenicity genes, which is possibly due to recombination and/or TEs dynamics. To avoid misinterpretation, we have reworded this part.

COMMENT: *Line 261-263, is the signal for positive selection only in the signal peptides? I ask because signal peptides don't have a specific sequence but instead a general physical property on which their function relies. As such, the synonymous/nonsynonymous ratio in these is typically greater than for a typical protein (i.e. TF or enzyme) where the specific sequence and structure have a tighter relationship. I'm also a bit puzzled as to why the signal peptide would be under diversifying selection? Or am I misreading and it is the whole protein that is showing this signal?*

RESPONSE: It is the mutations on these proteins that show signatures of positive selection but not the signal peptide (which is generally well-conserved in the vast majority of the analyzed sequences). The point we were trying to make with this statement is, the four genes that show the strongest signatures of positive selection do have a predicted signal peptide. However, to avoid the confusion we changed the phrase "We also found four genes with a predicted signal peptide having a $dN/dS > 1$ " to "We also found four genes with a $dN/dS > 1$ ".

COMMENT: *Line 172, I'm having a bit of trouble with the average phenotypic variance being >9% in all cases given the number of loci found per trait. Is the total variance adding up to >100% for a trait. This might arise from the way the section starts in line 155 where all the traits GWA are described across all traits but it isn't quite clear in the text how many traits this is actually representing and how many SNPs are found per trait. This is somewhat shown in Figure 2 but the text could be helped by representing this.*

RESPONSE: The phenotypic variance reported here is the one explained by the significant SNPs associated with the two pathogenicity traits, on each of the 12 cultivars. It is important to mention that association analyses were conducted independently for every trait/cultivar combination (e.g., PLACN trait on cultivar "Cadenza"), meaning that the reported R^2 values are specific to these combinations and cannot be added up. As suggested, we have clarified those details in the text to help in interpreting these figures.

COMMENT: *Line 173, is $R^2 < 30\%$ or absolute 30?*

RESPONSE: It was an error from our side and we have now corrected it to $R^2 < 30\%$.

Reviewer #2 (Remarks to the Author):

COMMENT: *The goal of this manuscript was to use genomic sequencing of 103 isolates of Zymoseptoria tritici in a GWAS analysis to identify quantitative pathogenicity. 58 candidate genes were identified and expression analysis was used to identify those with likely effects. Most showed signatures of diversifying selection and two were cloned for functional analysis. A conclusion was that many of the QTL likely showed specific interactions with host cultivars and had differing modes of action.*

The manuscript overall was very well done and the conclusions were sound. It could be improved by adding hypothesis testing to the last paragraph of the Introduction so that the reader can know exactly which knowledge gaps are being addressed and how the results advance our understanding. Lines 111-118 provided a nice summary of what was done, but it would be much better to provide one or more specific, scientific hypotheses that can be tested with the data. This will be easy to add as there are several obvious possibilities. Testing of clear hypotheses will provide greater clarity about exactly what you have done and how it moves the science forward.

RESPONSE: We thank you very much for this positive feedback. As suggested, we have added the specific hypotheses we are testing and the knowledge gaps we are addressing in the last paragraph of the introduction.

COMMENT: *In lines 402-406 the authors might want to look up Front. Microbiol. 2019; 10: 2210, in which the velvet gene was knocked out in Z. tritici.*

RESPONSE: Thank you for providing the reference on the velvet gene *ztve1B*. We have now included this reference and a sentence highlighting the results from this related work in the discussion.

COMMENT: *The References have some variability in formatting – scientific names not italicized, all words capitalized – that should be corrected. I circled those on the first two pages but the authors can check after that.*

RESPONSE: We have now formatted all the references according to the standard format of Nature Communications.

COMMENT: *To save time typing, I wrote out most of my comments – which in general were very minor – on a hard copy of the manuscript and will return them in a scan. Hope you can read my writing. Very nice job overall.*

RESPONSE: We appreciate the written feedback given throughout the manuscript. We have corrected the text and addressed all the comments accordingly. There is a single exception to this regarding the increase in Font size of gene labels in Figure 2. In fact, we have done so to the maximum readable. Further increase will result in a label overlap, which will make them unreadable (especially in some chromosomes where the detected genes are very close to each other; e.g., Chr2, Chr8, Chr9 etc.). We apologize for this inconvenience but some options for the readers will be to zoom in the figure, or to look at the supplementary table 4, where we provide those details.

REVIEWERS' COMMENTS

Reviewer #1 (Remarks to the Author):

The authors have nicely responded to the majority of my concerns. I only have three areas that are editorial that remain a concern.

Structural variance – There is evidence from multiple groups that read depth beyond a surprisingly minimal level does not help to identify structural variants as the issue is in the aligners ability to find these. I agree the 30x level is sufficient for SNPs at loci without structural variants but there is really no short-read solution at loci with structural variation. I would suggest providing the reader some discussion on these points. For example, the information on identifying only a single missing gene from the reference genome is a somewhat unique situation in fungal pathogens and merits reporting. These caveats about how the methodology constrains the analysis are important in providing the reader the ability to assess how approximate the image is for this genome.

I totally agree that quantitative genetics in fungal plant pathogens remains relatively unexplored but does that mean that no citations for these other systems should be provided? Presently the way the paragraph on line 66-81 is written, there are no studies being done on quantitative variation in fungal pathogens or fungal plant pathogens. Simply providing some citations to other systems in the sentence that ends on line 78 would give the reader the ability to assess how this manuscript fits into the broader community's research. At present the manuscript is written as if there is no community and this species is the only one with information to consider or discuss. Given the findings in these others systems, they would support the main ideas in the discussion but by not citing, this support is absent.

In Figure 5 and 6, are the rectangles in panel A separate genes or separate exons for a given gene? I think this was part of my concern about LD in significant regions.

REVIEWERS' COMMENTS

Reviewer #1 (Remarks to the Author):

The authors have nicely responded to the majority of my concerns. I only have three areas that are editorial that remain a concern.

Structural variance – There is evidence from multiple groups that read depth beyond a surprisingly minimal level does not help to identify structural variants as the issue is in the aligners ability to find these. I agree the 30x level is sufficient for SNPs at loci without structural variants but there is really no short-read solution at loci with structural variation. I would suggest providing the reader some discussion on these points. For example, the information on identifying only a single missing gene from the reference genome is a somewhat unique situation in fungal pathogens and merits reporting. These caveats about how the methodology constrains the analysis are important in providing the reader the ability to assess how approximate the image is for this genome.

I totally agree that quantitative genetics in fungal plant pathogens remains relatively unexplored but does that mean that no citations for these other systems should be provided? Presently the way the paragraph on line 66-81 is written, there are no studies being done on quantitative variation in fungal pathogens or fungal plant pathogens. Simply providing some citations to other systems in the sentence that ends on line 78 would give the reader the ability to assess how this manuscript fits into the broader community's research. At present the manuscript is written as if there is no community and this species is the only one with information to consider or discuss. Given the findings in these others systems, they would support the main ideas in the discussion but by not citing, this support is absent.

In Figure 5 and 6, are the rectangles in panel A separate genes or separate exons for a given gene? I think this was part of my concern about LD in significant regions.

Please, find below our response to Reviewer #1's remarks.

Reviewer #1 (Remarks to the Author):

The authors have nicely responded to the majority of my concerns. I only have three areas that are editorial that remain a concern.

Structural variance – There is evidence from multiple groups that read depth beyond a surprisingly minimal level does not help to identify structural variants as the issue is in the aligners ability to find these. I agree the 30x level is sufficient for SNPs at loci without structural variants but there is really no short-read solution at loci with structural variation. I would suggest providing the reader some discussion on these points. For example, the information on identifying only a single missing gene from the reference genome is a somewhat unique situation in fungal pathogens and merits reporting. These caveats about how the methodology constrains the analysis are important in providing the reader the ability to assess how approximate the image is for this genome.

To address the reviewer's concern, we added the following sentence in the discussion (lines 451-454):

“However, it is important to note that the utilization of short-read sequencing technology imposes limitations on our ability to identify structural variants that may play a role in pathogenicity. Genes exhibiting presence/absence polymorphism could be underrepresented in our final set of candidate pathogenicity genes.”

I totally agree that quantitative genetics in fungal plant pathogens remains relatively unexplored but does that mean that no citations for these other systems should be provided? Presently the way the paragraph on line 66-81 is written, there are no studies being done on quantitative variation in fungal pathogens or fungal plant pathogens. Simply providing some citations to other systems in the sentence that ends on line 78 would give the reader the ability to assess how this manuscript fits into the broader community's research. At present the manuscript is written as if there is no community and this species is the only one with information to consider or discuss. Given the findings in these others systems, they would support the main ideas in the discussion but by not citing, this support is absent.

In fact, quantitative variation in the pathogenicity of fungal plant pathogens are commonly measured and studied. However, our intention was to emphasize the limited availability of information regarding the genetic basis of these traits. In response to the reviewer's feedback, we have incorporated three additional references (lines 81-85) to underscore this point and fit our work into the broader community's research.

In Figure 5 and 6, are the rectangles in panel A separate genes or separate exons for a given gene? I think this was part of my concern about LD in significant regions.

This was explained in the legend of Figure 5: "Annotations on the reference genome are shown at the bottom with the candidate gene represented by a green box, other genes by blue boxes and transposable elements by grey boxes."

Thus, the rectangles in panel A of Figures 5 and 6 represent separate genes. We now completed the legend of Fig. 6 accordingly.